# Crystal Structure-Guided Design of Bisubstrate Inhibitors and Photoluminescent Probes for Protein Kinases of the PIM Family

**DOI:** 10.3390/molecules26144353

**Published:** 2021-07-19

**Authors:** Olivier E. Nonga, Darja Lavogina, Erki Enkvist, Katrin Kestav, Apirat Chaikuad, Sarah E. Dixon-Clarke, Alex N. Bullock, Sergei Kopanchuk, Taavi Ivan, Ramesh Ekambaram, Kaido Viht, Stefan Knapp, Asko Uri

**Affiliations:** 1Institute of Chemistry, University of Tartu, 14A Ravila St., 50411 Tartu, Estonia; olivier.etebe.nonga@ut.ee (O.E.N.); darja.lavogina@ut.ee (D.L.); erki.enkvist@ut.ee (E.E.); katrin.kestav@ut.ee (K.K.); sergei.kopanchuk@ut.ee (S.K.); taavi.ivan@ut.ee (T.I.); ramcheme@gmail.com (R.E.); Kaido.viht@ut.ee (K.V.); 2Institut für Pharmazeutische Chemie, Goethe University Frankfurt am Main, Max-von-Laue-Str. 9, 60438 Frankfurt am Main, Germany; chaikuad@pharmchem.uni-frankfurt.de (A.C.); knapp@pharmchem.uni-frankfurt.de (S.K.); 3Structural Genomics Consortium (SGC), Buchmann Institute for Life Sciences, Goethe University Frankfurt, Max-von-Laue-Str. 15, 60438 Frankfurt am Main, Germany; 4Centre for Medicines Discovery, Old Road Campus, University of Oxford, Roosevelt Drive, Oxford OX3 7DQ, UK; sarah.dixon-clarke@dfci.harvard.edu (S.E.D.-C.); alex.bullock@cmd.ox.ac.uk (A.N.B.); 5Department of Cancer Biology, Dana-Farber Cancer Institute, 450 Brookline Avenue, Boston, MA 02215, USA

**Keywords:** protein X-ray crystallography, co-crystal structure, PIM kinases, bisubstrate inhibitors, fluorescent probes, cellular uptake and localization, adenosine–arginine conjugate

## Abstract

We performed an X-ray crystallographic study of complexes of protein kinase PIM-1 with three inhibitors comprising an adenosine mimetic moiety, a linker, and a peptide-mimetic (d-Arg)_6_ fragment. Guided by the structural models, simplified chemical structures with a reduced number of polar groups and chiral centers were designed. The developed inhibitors retained low-nanomolar potency and possessed remarkable selectivity toward the PIM kinases. The new inhibitors were derivatized with biotin or fluorescent dye Cy5 and then applied for the detection of PIM kinases in biochemical solutions and in complex biological samples. The sandwich assay utilizing a PIM-2-selective detection antibody featured a low limit of quantification (44 pg of active recombinant PIM-2). Fluorescent probes were efficiently taken up by U2OS cells and showed a high extent of co-localization with PIM-1 fused with a fluorescent protein. Overall, the developed inhibitors and derivatives represent versatile chemical tools for studying PIM function in cellular systems in normal and disease physiology.

## 1. Introduction

The PIM (proviral integration site for Moloney murine leukemia virus) family of protein kinases (PKs) includes three constitutively active serine/threonine kinases (PIM-1, PIM-2, and PIM-3) that regulate key biological processes, including cell survival, proliferation, differentiation, and apoptosis [1,2]. Elevated expression levels of PIM kinases have been observed in hematologic malignancies such as myelomas and non-Hodgkin lymphomas [3,4]. PIM kinases play important roles in the development and progression of other types of cancer (e.g., non-small-cell lung cancer, pancreatic and prostate cancers, gastric, hepatocellular and squamous cell carcinomas, liposarcoma, glioblastoma) [2,5]. These findings suggest that PIM kinases are potential cancer drug targets and biomarkers [6]. Recent success in the development of PIM-selective inhibitors with low picomolar inhibitory potency [4] has intensified clinical testing of these inhibitors for the treatment of hematologic cancers.

Protein kinases (PKs) have been the most important targets for cancer drug development in this century. More than 60 small-molecule PK inhibitors have reached the drug market [7]. Most of these inhibitors follow Lipinski’s rules for oral drugs [8] and bind to the ATP-binding pocket or flanking regions of PKs. However, all PKs as well as other proteins of the purinome (a total of 3266 proteins encoded in the human genome) bind purine and its derivatives (e.g., adenine, the heteroaromatic moiety of ATP) [9]. Therefore, ATP-mimicking PK inhibitors bear a high off-target risk.

In recent years, the bisubstrate approach has gained popularity for the construction of potent and selective inhibitors of PKs [10,11,12]. Bisubstrate inhibitors consist of two conjugated fragments, each targeting the binding site of a particular substrate. This approach facilitates the formation of additional interactions between the inhibitor and less-conserved protein substrate-binding site of the kinase. Thus, the two important biochemical characteristics of an efficient inhibitor, affinity and selectivity, can be improved.

Previously, we have used the bisubstrate approach for the development of high-affinity bisubstrate inhibitors, ARC inhibitors (ARC: adenosine–arginine conjugate) possessing values of the dissociation constant (*K_D_*) in the picomolar range for several PKs: catalytic subunit α of protein kinase A (PKAcα), PIM-1, Rho-associated protein kinase 2 (ROCK2), and Haspin [13,14,15,16]. In ARCs, an adenosine analogue heteroaromatic moiety and substrate peptide mimetic d-Arg-rich fragment are covalently tethered by a linker. These conjugates are proteolytically stable and efficiently penetrate the plasma membrane of mammalian cells [17,18]. The latter property of ARCs is based upon the transport peptide fragment (d-Arg)_6_ incorporated into several ARC inhibitors [17]. In case of basophilic PKs, the (d-Arg)_6_ fragment associates with the substrate protein-binding domain of the target PK and thereby substantially contributes to the binding of ARCs to the PK. On the other hand, oligo-(d-Arg) peptides comprising six or more d-Arg residues are considered as classical transport peptides that efficiently penetrate the cell plasma membrane. Unfortunately, (d-Arg)_6_ peptide is highly charged and can develop specific and non-specific interactions with certain biomolecules (e.g., endoprotease furin (*K_D_* = 106 nM and 1.3 nM for (d-Arg)_6_-NH_2_ and (d-Arg)_9_-NH_2_, respectively) [19], nucleic acids [20], components of cell nucleoli [21]) as well as labware surfaces [22].

Here, we report co-crystal structures of ARC/PIM-1 complexes. Based on the newly developed ARC-type inhibitor BPTP-Ahx-(d-Arg)_6_-d-Lys-NH_2_ (ARC-3126; BPTP-7-bromo-2-(methylene)pyrido[4,5]thieno[3,2-d]pyrimidin-4-one, Ahx-6-aminohexanoic acid), new compounds with a reduced number of d-Arg residues were constructed in order to decrease the risk of non-specific interactions in biochemical experiments. The affinities of the novel compounds were assessed along with their selectivity profiles. An ARC-affinity surface was designed for capturing PIM-2, which was quantitatively detected with a specific monoclonal antibody. Finally, fluorescent probes derived from the newly developed compounds were examined in live U2OS cells to assess their cell plasma membrane-penetrative properties and intracellular localization.

## 2. Results and Discussion

### 2.1. Thermal Shift Assay of ARC/PIM Complexes

The study was started by choosing a set of structurally diverse ARCs, which had previously revealed low nanomolar or subnanomolar *K_D_* values toward their reported PK targets (PKAcα, ROCK2 etc.) [13,23]. For establishing affinity toward different PKs of the PIM family, a thermal shift assay was used that measured the stabilization of the 3D structure of the PK upon its binding to the compound under study [24]. As control compounds, an ATP-binding site PIM inhibitor SGI-1776, as well as ARC-3119 and ARC-3125 (previously tested with PIM kinases) were utilized.

The results of measurements are summarized in Table 1. All the characterized compounds exhibited Δ*T*_m_ values of over 5 °C with all PIM isoforms (according to the literature, Δ*T*_m_ values over 5 °C typically point to nanomolar affinity of inhibitors [24]). SGI-1776 featured a Δ*T*_m_ value of 9.5 °C with PIM-1, which is comparable with the previously reported data (Δ*T*_m_ value of 9.7 °C was reported for a very similar structural analogue of SGI-1776 [25]).

All tested ARCs featuring large Δ*T*_m_ values contained at least six d-Arg residues, whereas the binding of compounds incorporating TIBI ((4,5,6,7-tetraiodo-1H-benzimidazol-1-yl)acetic acid moiety) and BBTP (8-bromo-2-(methylene)[1]benzothieno[3,2-d]pyrimidin-4-one moiety) as adenosine analogue moieties led to higher stability of the protein structure. In our previous study, ARCs with AMTH and PIPY bound with high affinity to a large set of basophilic PKs [13,23], whereas the BBTP moiety conferred a PIM-selective inhibitor [14] and the TIBI moiety conferred a promiscuous CK2 inhibitor [26]. The TIBI-comprising conjugate ARC-3125 revealed high Δ*T*_m_ values (12.7–14.7 °C) due to the strong binding of both its moieties [26] to the substrate pockets of PIM. Moreover, ARC-3125 stabilized PIM kinases significantly more compared to previously studied PKs DYRK2 and CLK2 that showed Δ*T*_m_ values of only 3.3 °C and 4.0 °C, respectively [15].

### 2.2. Co-Crystal Structures of ARC-1411, ARC-1415, and ARC-3126 with PIM-1

Based on the results of thermal shift assay, we selected the candidates for co-crystallization with PIM-1: ARC-1411, ARC-3119, and their derivatives ARC-1415 and ARC-3126. Adenosine analogue moieties of ARC-1415 and ARC-3126 were structurally different (PIPY vs. BPTP, respectively), and these two compounds also comprised different linkers (α,ω-nonanedioic acid vs. 6-aminohexanoic acid). The only structural difference between ARC-1415 and ARC-1411 was the lack of a C-terminal d-Lys residue in ARC-1415 (Figure 1).

Unfortunately, the co-crystallization of ARC-3119 with PIM-1 was not successful; co-crystals were obtained for PIM-1 complexes with ARC-1411, ARC-1415, and ARC-3126. The corresponding crystal structures were solved and refined at 1.9 Å resolution (Appendix A) and the coordinates were deposited in the protein databank (PDB) (ID 7OOV for ARC-1411, 7OOW for ARC-1415, and 7OOX for ARC-3126). All crystal structures contained one protein molecule in the asymmetric unit, in which the adenosine analogue moiety of the inhibitors was positioned in the ATP-binding site of PIM-1 and the peptide mimetic moiety bound on the surface of the substrate protein site (Figure 2). The electron densities for the ARCs in the obtained co-crystals are shown in Appendix A. The peptide mimetic moiety of ARC-1411 was resolved in the complex with PIM-1 to a larger extent compared to the previously reported complex with PKAcα [13]. However, some of the side chains of the peptidic portion of the inhibitor that were oriented toward the solvent showed weak electron density. The same applied to the ARC-1415 complex, where first d-Arg (d-Arg1; Figure 1) and a small portion of the linker as well as the side chain of d-Arg5 were poorly resolved; again, these moieties made few interactions with the PIM-1 catalytic domain. In contrast, in the ARC-3126 complex, the linker and the side chain of d-Arg1 were well resolved, but the terminal arginine and lysine residues showed only weak density in the 2F_o_-F_c_ electron density maps (Appendix A). As the side chains of the terminal arginine and lysine made few interactions, the peptide portion of these bisubstrate inhibitors could be truncated at d-Arg5 without losing affinity for the target.

The structure of the ARC-1411 complex with PIM-1 revealed a canonical type-1 inhibitor interaction of its pyrrolo-pyrimidine hinge binding moiety, forming the anticipated hydrogen bond of the pyrrolo nitrogen with the backbone carbonyl of the hinge residue E121. Due to the insertion of a proline residue in the hinge of PIM kinases, the pyrimidine moiety did not engage in additional hinge backbone hydrogen bonds. Comparison with a complex of the PIM-1 kinase with a consensus peptide (PDB-ID: 2BIL) revealed high conservation of key interactions with consensus substrate residues (Figure 2). As a basophilic kinase of the CAMK family, PIM kinases recognize basic residues, in particular arginines at positions -3 and -5 of the phosphorylation site, which are required for high affinity interaction [27] (Figure 2B,C).

These key interactions were closely mimicked by arginine moieties in ARC-1411. Remarkably, the first arginine moiety connected via an aliphatic linker to the ATP mimetic hinge binding motif engages with the glycine-rich loop (GRL) residue S46. The GRL itself assumed a folded conformation, which is often observed in PIM-1 inhibitor complexes where the phenylalanine (F49) is positioned at the tip of the GRL and flipped into the ATP binding pocket [28]. The second arginine in ARC-1411 mimicked the interaction of the arginine in the key substrate position -3, which pointed into an acidic binding pocket lined by the αD residues D128 and D131. The following arginine in ARC-1411 formed polar interactions with D239 that were not present in the substrate complex, most likely increasing the affinity of ARC-1411 (Figure 2C). However, similar to the substrate residue R-4, the next arginine moiety in ARC-1411 pointed toward the solvent, but the following arginine mimicked again the substrate arginine in position -5, which is required for high-affinity substrate interactions. At this position, the arginine side chains formed an intricate network of polar interactions with residues located in αD (T134), αF (D234), and D170. The terminal arginines in ARC-1411 formed a number of long-range salt bridge (D243) and stacking interactions (R256). In summary, the well-resolved ARC-1411 complex with PIM-1 revealed the presence of key interactions that are critical for high-affinity substrate binding as well as additional interactions that explain the high affinity of this bisubstrate inhibitor for PIM kinases. Similar interactions were also conserved in the closely related ARC-1415 complex (Figure 3A).

In contrast, ARC-3126 utilized a different exit vector based on its diverse ATP-competitive moiety BPTP (Figure 3C,D). Remarkably, ARC-3126 did not form canonical hydrogen bonds with the hinge backbone but was anchored to the backpocket via a polar interaction with the conserved VAIK lysine (K67) similar to other kinases that harbor a large hydrophobic residue immediately N-terminal to the DFG motif [29], which is a binding mode that has been associated with favorable selectivity profiles (Figure 3D). The non-classical hinge-binding mode of BPTP in the ARC-3126/PIM-1 complex, similar to those of pyrimidinones and pyridazines [25,30], is a distinctive element for its selectivity, and thus for the selectivity of ARC-3126 over the promiscuous inhibitor ARC-3125. Consistently, ARC incorporating a pyrimidinone derivative as an ATP mimetic moiety possessed higher affinity toward PIM kinases than to other basophilic PKs in a previously reported selectivity panel [14], whereas ARC-1411 and ARC-1415 were reported to inhibit a wide list of basophilic PKs [13]. Attachment of the linker to the peptidic portion of ARC-3126 at the 2 position of the pyrimidin-4-one ring resulted in reorientation of the GRL, forcing F49 into an extended active-like conformation (Figure 3C,D).

In the ARC-1411 and ARC-1415 co-crystals, the d-Arg1 interacted with the GRL; in the case of ARC-3126, the interaction with GRL was formed by the carboxyl group of the 6-aminohexanoic acid linker (Figure 3A–C). Except for d-Arg1, the peptide mimetic moieties of ARC-1411, ARC-1415, and ARC-3126 formed similar polar interactions with PIM-1. Remarkably, despite the different exit vectors, the main substrate mimetic interactions with d-Arg2, d-Arg3, and d-Arg5 were highly conserved and superimposed well with the peptidic portion in ARC-1411. d-Arg2 interacted with PIM-1 residues D128 and D131, d-Arg3 with D239, and d-Arg5 with F130, D170, and T134. Additionally, d-Arg5 of ARC-1411 and ARC-1415 made contacts with D234 and G238. On the other hand, the C-terminus of the inhibitor showed a high degree of structural diversity, and d-Arg4 and d-Arg6 did not interact with the protein (Figure 3A–C).

PIM-1 has a strong preference for substrates with basic residues, particularly Arg, at positions −5 and −3 from the phosphorylatable serine or threonine residue [25,27]; for instance, the PIM-1 substrate peptide Pimtide (ARKRRRHPSGPPTA) possesses very high affinity to the kinase (*K_D_* value of 58 nM [27]). According to the previously reported crystal structure (PDB-ID: 2BZK, [27]), Arg residues at −5 and −3 positions in Pimtide interact with the same PIM-1 residues as ARC-1411, ARC-1415, and ARC-3126. Therefore, ARC inhibitors that were the focus of this study have the capacity of preventing interactions involving PIM kinases and their substrate proteins, similarly to a structurally different inhibitor that also reveals multiple interactions with residues of the substrate-binding site [31].

### 2.3. Design and Biochemical Characterization of New Inhibitors

Guided by the 3D structure of the PIM-1 bound lead compound ARC-3126 and taking into consideration the revealed cues regarding the putative binding mode and selectivity features of the compound, a new set of ARC compounds was designed. As the crystal structure analysis revealed that d-Arg1, d-Arg4, and d-Arg6 residues of ARC-3126 did not contribute to the binding in the ARC-3126/PIM-1 complex, these structural elements were omitted from the new ARC inhibitors to make the compounds structurally smaller, simpler, and less charged. To conserve the positioning of d-Arg2, d-Arg3, and d-Arg5, longer linker chains were used in inhibitors, and the d-Arg4 residue was replaced by Gly. The C-terminal d-Lys residue was kept for potential labeling of inhibitors via the amino group of its side chain. Potentially, these new structures were expected to reveal lower non-specific binding to components of biological solutions, as well as to become less prone to binding to nucleic acids in the cellular context [17].

The affinities of the compounds toward the PIM family PKs were determined in a binding/displacement assay with measurement of fluorescence anisotropy (FA) and time-gated luminescence intensity (TGLI) (Table 2). For the initial assessment of the selectivity of compounds, PKAcα was used as the reference basophilic kinase [10]. As a control compound, the commercially available ATP-competitive PIM inhibitor AZD1208 was used.

The six d-Arg residues-comprising lead compound ARC-3126 possessed a *K_D_* value of 1.8 nM toward PIM-1 and, in comparison, a *K_D_* value of over 18 μM toward PKAcα. Compound **1** (incorporating an 8-aminooctanoic acid (Aoc) linker) bound to PIM-1 with a *K_D_* value of 11.4 nM but was more than 50-fold less selective than ARC-3126 based on comparison of their respective affinities toward PIM-1 and PKAcα (Table 2). This may reflect the fact that compound **1** did not interact with all the substrate binding hotspots of PIM-1 compared to ARC-3126. Upon replacement of the adenosine analogue moiety BPTP (compound **1**) with BBTP (compound **2**) that has been reported to have high affinity and selectivity to PIM kinases [30], the selectivity to PIM-1 was increased 11-fold.

PKAcα binds with high affinity to ARCs that comprise a flexible linker between the adenosine analogue and peptide mimetic moieties [13]. Aoc is a highly flexible linker in compounds **1** and **2**. Therefore, rigidified linkers comprising a 1,2,3-triazole fragment were tested in compounds **3** and **4**. The 1,2,3-triazole moiety is resistant to cellular proteolytic degradation and capable of forming hydrogen bonds. When [4-(aminomethyl)-1H-1,2,3-triazol-1-yl]acetic acid (Amt) was used as the linker, the affinity and selectivity of the resulting compound **3** was comparable to those of compound **2** (Table 2). The use of another linker, 5-(1H-1,2,3-triazol-4-yl)pentanoic acid (Tap) in compound **4** led to over 5-fold higher selectivity as compared to compound **3**.

Furthermore, it has been shown that ARC inhibitors harboring a hydroxyproline (Hyp)-incorporating linker possess good PIM selectivity [14]. Therefore, **2** was modified by the insertion of Hyp between BBTP and Aoc. This modification resulted in compound **5** possessing over 4-fold lower selectivity toward PIM-1 than compound **2**, suggesting less suitable positioning of the peptide mimetic moiety of **5** in complex with PIM-1.

Fluorescent probes are important tools in biochemical research due to the high sensitivity of fluorescence-based assays, applicability of the probes for high-throughput measurements, and amenableness of the assays for automation [34]. The labeling of compounds **2** and **4** with the fluorescent cyanine dye Cy5 via the side-chain amino group of the C-terminal d-Lys residues yielded fluorescent probes **6** and **7**, featuring nanomolar affinity toward PIM-1 (*K_D_* = 1.7 nM and 21.9 nM, respectively). The *K_D_* value of **6** demonstrated that attachment of the dye did not affect much the binding of the inhibitor to the kinase and pointed to the potential applicability of the probe for screening of PIM inhibitors. Importantly, the affinities of **6** and **7** to PKAcα were much lower, falling in the sub-micromolar and micromolar ranges (Table 2). The analogue of **7** with 5-TAMRA dye (compound **8**) revealed similar affinity to PIM-1 and PKAcα as **7**.

The binding properties of the designed peptide mimetic moiety of compounds **1–5** (PIM peptide) to PIM-1 were also tested. It revealed only weak affinity, possessing a *K_D_* value of over 10 µM (Table 2). Finally, the tested compounds revealed almost the same affinity and selectivity to PIM-3 compared to PIM-1, while they demonstrated lower affinity to PIM-2 (except for ARC-3126, which showed similar affinity toward all PIM kinases; Table 2).

### 2.4. Kinome-Wide Selectivity Profiling

To evaluate the kinome-wide selectivity of novel ARCs, compounds **1**, **2**, and **8** as well as the lead compounds ARC-3125 and ARC-3126 were tested against a panel of 140 PKs in the MRC PPU International Centre for Kinase Profiling in Dundee.

The residual activities of PKs in the presence of 1 µM final concentration of compounds are summarized in Appendix A. The profiling confirmed the promiscuous behavior of TIBI-comprising ARC-3125 with a Gini coefficient of 0.508 and a hit rate of 0.62 (the latter value indicated that 62% of the tested PKs had residual activity below 50%). The PKs with residual activity below 20% belonged to different groups of the kinome [35]. As expected, high inhibitory potency was revealed toward the PIM family (CAMK group), yet the residual activity of another PK of the CMGC group CDK1/cyclin A, a STE group member MAP4K3, a tyrosine kinase-like group member IRAK1, as well as the tyrosine kinase IGF1R were all below 25% residual activity at 1 µM inhibitor concentration.

The selectivity of ARC-3126 was highlighted by a Gini coefficient value of 0.73 and a hit rate of 0.09 (Appendix A). Notably, PIM-1 had zero residual activity, while PIM-2 and PIM-3 revealed residual activity of 10% and 13%, respectively (Appendix A). The major off-target of ARC-3126 was MAPKAP-K2 kinase with a residual activity of 13%; the other members of the MAPKAP family were not inhibited. The observed inhibition of CK2 in the kinase panel was most likely an assay artefact caused by the interaction of Arg-rich ARCs with the aspartic acid-rich substrate peptide of CK2.

On the other hand, compound **1** comprising three d-Arg residues showed a remarkably narrow selectivity profile compared to ARC-3126, with a Gini coefficient value of 0.887 and hit rate of 0.022 (Appendix A). Compound **1** strongly inhibited kinases of the PIM family (residual activities of 9% and 17% for PIM-1 and PIM-3, respectively) except for PIM-2, which unexpectedly retained 72% of residual activity. Other basophilic kinases in the panel had high residual activities above 81%.

Despite the higher affinity of compound **2** toward PIM kinases compared to compound **1** (Table 2), the selectivity profile of **2** was slightly broader, as revealed by its Gini coefficient value of 0.824 and a hit rate of 0.079 (Appendix A). Still, these parameters indicated that the selectivity profile of **2** was narrower than that of our previously reported selective CK2 inhibitor ARC-772 (Gini coefficient of 0.752, [36]), or that of the clinically used Bcr-Abl inhibitor imatinib (Gini coefficient of 0.76, [37]). Expectedly, compound **2** strongly inhibited PKs of the PIM family (residual activity in the range of 4–7% for all isoforms). The major off-target PKs of **2** compared to **1** were DYRK1α (residual activity of 5% vs. 74%), CLK2 (24% vs. 97%), and RSK1 (31% vs. 95%).

The radiometric assay format used for the kinase panel also enabled the screening of fluorescent molecules. 5-TAMRA-labeled compound **8** displayed a Gini coefficient of 0.682 and hit rate of 0.066 (Appendix A); in this respect, **2** and **8** demonstrated a similar level of selectivity and possessed similar ratios of the Gini coefficient over hit rate (around 10). Accordingly, **8** inhibited the PIM family to the same extent as **2**. The examples of kinases inhibited less potently by **8** as compared to **2** included DYRK1α (residual activity of 37% vs. 5%), RSK1 (74% vs. 31%), and TAO1 (80% vs. 49%) kinases. Contrarily, some kinases were inhibited more potently by **8** as compared to **2**, such as PKBβ (residual activity of 54% vs. 101%), TBK1 (46% vs. 113%), and CAMK1 (49% vs. 81%).

### 2.5. Surface Sandwich Assay for the Measurement of PIM-2 Concentration

PIM-2 is largely expressed in both leukemia and solid tumors. It plays an important role in promoting cell survival and preventing apoptosis [38]. Therefore, new analytical methods are needed to assess its expression level and activity in cells, cell lysates, and tissue extracts.

Previously, we have worked out a method for PIM-2 analysis and screening of its inhibitors in cell lysates [39], yet that method required the utilization of fluorescently tagged PIM-2 protein, and it could not be used for measurement of subnanomolar concentrations of the native kinase. Thus, we aimed at the development of a more sensitive and more robust method for the quantitative detection of PIM-2 with the aid of new ARC-probes.

To our knowledge, there are few reports on protein analysis assays based on the formation of a three-component complex [40] on solid support that uses the combination of a small-molecule inhibitor and specific antibody as capture/detection reagents [41]. Usually, larger molecules (proteins, nucleic acids, aptamers, etc.) are used for establishing such assays [42]. For example, PIM-2 can be analyzed with conventional enzyme-linked immunosorbent assays (ELISA). On the market, there are various sandwich-type ELISA kits from several suppliers. On the other hand, a classical ELISA assay requires the application of two orthogonal antibodies of the analyte, which are not always easy to obtain [43]. In this study, we have substituted an ARC-inhibitor-based capture molecule for the capture antibody. ARC inhibitors are synthetic mid-sized organic molecules that feature several advantages as compared to an antibody, such as shorter generation time, lower costs of manufacturing, the lack of batch-to-batch variability, higher modifiability, and better thermal stability. Another distinctive feature of the ARC-utilizing assay that might be physiologically relevant is that only the active kinase is captured, whereas the non-active (e.g., denatured) PIM-2 is washed out.

For the surface assay, ARC-2073, a biotinylated derivative of compound **2** was synthesized (Appendix A). ARC-2073 revealed the *K_D_* value of 10 ± 2 nM in solution toward PIM-2, which was comparable to that of compound **2** (Table 2). Following the immobilization of ARC-2073 onto a streptavidin surface, PIM-2 was captured from a solution containing the recombinant protein. Thereafter, the captured PIM-2 was detected using rabbit anti-PIM-2 antibody D1D2 and visualized with an europium chelate-labeled secondary goat antibody against rabbit immunoglobulin G (G0506) (Figure 4A). The obtained calibration graph is depicted in Figure 4B. The lowest quantifiable concentration of PIM-2 was 25 ± 1 pM (quantification range from 25 to 625 pM), pointing to the high sensitivity of the assay, affording the measurement (LoQ) of 1.3 fmol (44 pg) of PIM-2 in the sample (50 μL). Thus, this sensitivity is advantageous versus the PIM-2 ELISA kit from ELK Biotechnology (product #ELK6492), which has a quantification limit of 1.8 fmol.

To assess the selectivity of the assay, the ARC-2073-coated surface was treated with solutions of PIM-1, PIM-3, and DYRK1α (an off-target PK of ARC-2073). These kinases were not detected with the pair of detection reagents D1D2/G0506 (Appendix A), confirming that D1D2 is specifically binding to PIM-2. In essence, the developed assay can be used for screening antibodies orthogonal to ARC-2073. On the other hand, the developed method can be potentially adapted for the quantitative detection of PIM-1 or PIM-3 by switching to the appropriate primary antibodies.

According to the literature, D1D2 has been used for the specific detection of PIM-2 in cell lysates [44]. Unfortunately, our assay did not reliably work in complex biological solutions such as crude cell lysates (Appendix A), which was probably due to matrix effects (presence of other biological molecules) on the stability of ARC-2073/PIM-2 complex in such solutions. We are currently working on reversing the assay format to capture PIM-2 with a surface-immobilized antibody to overcome the aforementioned issues.

### 2.6. Cell Plasma Membrane-Penetrative Properties of Dye-Labeled ARCs

Fluorescence sensors have gained great importance in biomedical research. Fluorescent probes based on selective inhibitors can be helpful for mapping the localization, concentration, and activity of target proteins in cells [45].

Several fluorescent probes based on inhibitors binding to the ATP-binding site have been constructed for PKs of the PIM family. First, a fluorescent dansyl derivative of a pyrrolo[2,3-*a*]carbazole-based PIM-1 inhibitor (fluorescent probe PCC-13) was reported to penetrate the plasma membrane and locate to the cytoplasm in live cells [46]. Later, a red-emitting fluorescent probe NB-BF targeting PIM-1 kinase was developed for the imaging of cancer cells [47]. NB-BF was prepared by conjugating 5-bromobenzofuran-2-carboxylic acid (a PIM-1 inhibitor possessing an *IC_50_* value of 8.5 μM) with a red-emitting fluorophore Nile blue via hexanediamine linker. Unfortunately, the probe was not characterized in biochemical assays, and therefore, its binding affinity toward PIM-1 and selectivity to other PKs is unknown. The experiments performed in living cells and animals pointed to localization of the probe as well as PIM-1 protein in mitochondria. Recently, a novel near-infrared imaging agent QCAi-Cy7d was constructed by conjugating a small molecule inhibitor of PIM-1 and near-infrared heptamethine cyanine dye [48]. The effective antitumor activity of the novel theranostic agent was attributed to PIM-1 inhibition and mitochondrial targeting.

Fluorescently labeled ARCs comprising multiple d-Arg residues have been previously shown to penetrate the plasma membrane of live cells [17,49]. The uptake efficiency of ARCs depends on the structure of the adenosine analogue fragment, the number of d-Arg residues in the conjugate, and the hydrophobicity of the fluorescent dye. While the exact mechanism of the cellular entry of cell-penetrating peptides (CPPs) is still disputable, two main contributing pathways have been proposed—endosomal internalization and direct cell membrane penetration [50,51]. According to the mainstream paradigm, CPPs should comprise at least six guanidinium groups to demonstrate good penetrability of cell plasma membrane [51,52]. Previously, we reported that an ARC-probe comprising two d-Arg residues and a positively charged 5-TAMRA dye did not efficiently internalize into cells [17].

New PIM-targeting ARCs comprise three d-Arg residues; thus, we were interested in whether the derivatization of compounds **2** and **4** with a hydrophobic Cy5 dye, bearing a single positive charge, would facilitate the internalization of compounds **6** and **7** into cells. Due to its optical properties (excitation wavelength above 600 nm), Cy5 dye features reduced background cell autofluorescence in microscopy. As control compounds, compound **8** and ARC-2090 were used: the former compound contains three d-Arg residues and is labeled with TAMRA dye, whereas the latter compound contains six d-Arg residues and is labeled with Cy5 (Appendix A).

The ability of fluorescent compounds to cross the plasma membrane of live U2OS cells was examined by following the effects of the incubation time (10–60 min) and concentration of the compounds (1–5 μM). Remarkably, **6** and **7** efficiently internalized into the cells (Figure 5), whereas the fluorescence of cells with **6** was somewhat stronger. In line with previous studies [36], the cellular penetrability was strongly concentration-dependent, with uptake at 2.5 μM or higher concentrations being remarkably more efficient than at 1 μM concentration. At 5 μM probe concentration and after 60 min incubation, the concentration of the compound in cells was so high that dark vesicles could be seen in brightfield images of **6** (Appendix A). In contrast, **8** featured much lower internalization efficiency and gave a reduced signal-to-noise ratio in microscopy images. The higher background signal was mostly caused by adsorption of the probe to the surface of the plastic well (Appendix A). Compounds **2** and **4** labeled with a sulfonated analogue of Cy5 (sulfo-Cy5) could not penetrate the plasma membrane of U2OS cells even at 5 μM concentration (data not shown), which can be attributed to the negative charges of sulfo-Cy5, leading to repulsion from the negatively charged cell plasma membrane.

In cells, compounds **6** and **7** showed mainly perinuclear vesicular localization, with some additional cytoplasmic signal. As compared to ARC-2090 incorporating six d-Arg residues, the nuclear staining of compounds incorporating three d-Arg residues was remarkably lower (Appendix A), as it was conceived when the probes were designed.

To examine whether the vesicular staining of novel ARC probes might reflect the intracellular localization of PIM kinases or the compartmentalization is rather related to endosomatic uptake characteristic for many CPPs [53], we proceeded with co-localization studies using high-resolution total internal reflection-fluorescence (TIRF) microscopy. U2OS cells were transiently transfected for 48 h with genetic constructs representing either PIM-1 fused with GFP and PIM-1 fused with TagRFP, or endosomal marker Endo-14 fused with fluorescent protein mCherry [39,54]. As a negative control for co-localization studies, peroxisomal marker (Peroximal Targeting Signal 1) fused with mCherry was used (this marker is sequestered into the well-defined vesicular structures, yet there is no evidence-based reason to associate the intracellular location of PIM-1 with peroxisomes). Thereafter, 1 h incubation with 2.5 μM or 5 μM solutions of compounds **6** or **7** was carried out, followed by live cell imaging. PIM-1 was preferred as a model representative of the PIM family due to its higher ATP *K_M_* value of 400 μM compared to ATP *K_M_* values of PIM-2 and PIM-3 (4 and 40 µM, respectively) [4]. Therefore, in the cellular environment, the binding of ARCs to PIM-1 is less affected by high intracellular concentrations (1–10 mM) of ATP than the binding to PIM-2 and PIM-3.

We observed that in U2OS cells overexpressing PIM-1 fused with a fluorescent protein, the kinase showed enrichment in the nucleoplasm, which was combined with a somewhat lower cytoplasmic signal, which is consistent with previous reports [44,55,56]. However, under low expression conditions, the signal of PIM-1 was observed in perinuclear vesicles; importantly, in cells with vesicular PIM-1 localization, a high extent of co-localization between ARC probes (either compound **6** or **7**) and PIM-1 fusions (either GFP-PIM-1 or PIM-1-TagRFP) was evident (Figure 6A–C and Figure 7A–F; Appendix A). Both ARC and labeled proteins were confined to the well-defined spot objects inside the cell cytoplasm. The co-localization extent was estimated with Manders’ coefficient (assessing overlap of objects segmented from the labeled protein channel inside the objects segmented from the ARC channel). The highest co-localization was observed for ARCs and PIM-1-TagRFP: for compound **6**, Manders’ coefficient was 0.72 ± 0.06 (N = 9), and for compound **7**, Manders’ coefficient was 0.43 ± 0.16 (N = 4). Additionally, some co-localization between ARC probes and mCherry-Endo-14 could be detected (Figure 7G–I), with the Manders’ coefficient value of 0.20 ± 0.03 (N = 5). Expectedly, the Manders’ coefficient values for the PIM-targeting ARC and peroxisome marker were very low (≤0.02), indicating absence of co-localization.

## 3. Materials and Methods

### 3.1. Materials

Chemicals and resins for synthesis were obtained from Iris Biotech GmbH (Marktredwitz, Germany), Novabiochem (Läufelfingen, Switzerland), AnaSpec (Fremont, CA, USA), ABCR (Karlsruhe, Germany), Acros (Geel, Belgium), Alfa Aesar (Kandel, Germany), Fluka (Buchs, Switzerland; Seelze, Germany), Macherey-Nagel (Bethlehem, PA, USA), and Fisher Scientific (Geel, Belgium; Loughborough, UK; Ward Hill, MA, USA; Hampton, NH, USA). SGI-1776 and AZD1208 were from Axon Medchem (Reston, VA, USA) and Cayman Chemicals (Ann Arbor, MI, USA), respectively. N-hydroxysuccinimide esters of 5-TAMRA and Cy5 were obtained from Sigma-Aldrich (St. Louis, MO, USA) and Lumiprobe GmbH (Hannover, Germany), respectively. ARC-1141, ARC-3125 [15], ARC-668 [57], ARC-684 [23], ARC-3119, ARC-1188 [14], ARC-1411, and ARC-1415 [13] were synthesized as previously described. ARC-583 was previously developed for PKAcα and other PKs [32]. PIM kinases (recombinant human proteins, full sequence) were produced as previously described [27,58]. PKAcα (recombinant human protein, full sequence) was obtained from Biaffin GmbH & Co KG (Kassel, Germany). Rabbit monoclonal anti-PIM-2 antibody, clone D1D2 (2.5 μM) was purchased from Cell Signaling Technology (#4730; Danvers, MA, USA). Europium chelate-labeled goat anti-rabbit IgG (6.7 μM) was obtained from Tokyo Chemical Industry (G0506; Tokyo, Japan). Human bone osteosarcoma cell line U2OS was obtained from The Leibniz Institute DSMZ—German Collection of Microorganisms and Cell Cultures GmbH (Braunschweig, Germany); human prostate adenocarcinoma cell line PC-3 (derived from metastatic site in bone) was from the American Type Culture Collection (Manassas, VA, USA). The solutions and growth medium components for the cell culture were obtained from the following sources: phosphate-buffered saline (PBS), charcoal-purified fetal bovine serum (ccFBS), L-glutamine, Dulbecco’s modified Eagle’s medium (DMEM)/Ham’s F12 medium without phenol red, McCoy’s 5A medium, DMEM high-glucose medium, fetal bovine serum (FBS)—Sigma-Aldrich (Steinheim, Germany); a mixture of penicillin, streptomycin, and amphotericin B—Capricorn (Ebsdorfergrund, Germany); imaging medium components (MEMO and Supplement A)—Cell Guidance Systems (Cambridge, UK). Plasmids of the markers of peroxisomes (mCherry-Peroxisomes-2) and endosomes (mCherry-Endo-14) were a gift from Michael Davidson via Addgene (plasmid #54520 and #55040, respectively). The plasmids encoding GFP-PIM-1 and PIM-1-TagRFP fusion proteins were constructed as previously reported [39]. The transfection reagents FuGENE^®^ HD and TurboFect™ were obtained from Promega Corporation (Madison, WI, USA) and ThermoScientific (R0531; Vilnius, Lithuania), respectively.

### 3.2. Synthesis of Compounds

BBTP and BPTP were synthesized according to the reported procedure [14]. ARC-3126, compounds **1–5**, PIM peptide, and the precursors of ARC-1451 were synthesized on the Rink amide 4-methylbenzhydrylamine resin according to fluorenylmethoxycarbonyl (Fmoc) solid-phase peptide synthesis method as published earlier [14,59]. ARC-1451, ARC-2090, and compounds **6** and **7** were obtained by labeling their precursors with corresponding fluorescent dyes following the published protocol [14]. The detailed synthetic procedures as well as high-resolution mass spectrometry (HRMS) and HPLC purity data are presented in the Appendix A.

After synthesis, the compounds were purified with Shimadzu Prominence LC Solution HPLC system (Columbia, MD, USA) using a Phenomenex Luna C18 reverse-phase column (5 μm, 25 cm × 0.46 cm; Torrance, CA, USA). A Thermo Electron LTQ Orbitrap (Langenselbold, Germany) mass spectrometer was used for the determination of exact masses of synthesized compounds. NMR spectra were recorded on a Bruker Avance-III 700 MHz (16.4 T; Bruker BioSpin GmbH, Rheinstetten, Germany). NanoDrop 2000c spectrophotometer (Thermo Scientific; Wilmington, DE, USA) was used for quantification of the compounds based on the following molar extinction coefficient (ε) values: BBTP (5700 M^−1^cm^−1^ at 343 nm), BPTP (5400 M^−1^cm^−1^ at 308 nm), 5-TAMRA (80,000 M^−1^cm^−1^ at 558 nm), or Cy5 (250,000 M^−1^cm^−1^ at 646 nm). The ε value for PIM peptide was calculated according to the literature [60] as 5906 M^−1^cm^−1^ at 214 nm.

### 3.3. Thermal Shift Assay for ARC/PIM Complexes

Thermal shift assay measurements were performed following the previously published protocol [61] using a real-time PCR instrument (Mx3005p RT-PCR, Stratagene; La Jolla, California). The final total concentrations of PIM kinases and ARCs in the experiments were 2 and 12.5 μM, respectively. The measurements were carried out in PCR low-profile microplate wells (ABgene; Portsmouth, NH, USA). The temperature scan was run from 25 to 95 °C at 1 °C/min. GraphPad Prism v. 6.0 (GraphPad Software, Inc.; San Diego, CA, USA) and Microsoft Excel v. 2007 (Microsoft Office; Redmond, WA, USA) were used for data processing and analysis.

### 3.4. Co-Crystallization, Diffraction Data Collection, and Structure Determination

Final concentrations of PIM-1 and the inhibitors were 0.2 mM and 1 mM, respectively. Crystals of complexes were grown using the sitting-drop vapor-diffusion method at 4 °C (ARC-1411) or 20 °C (ARC-1415, ARC-3126) with a reservoir solution listed in Appendix A. Diffraction data were collected at Diamond Light Source, and processed and scaled with XDS [62] and AIMLESS [63], respectively. Structures were solved by molecular replacement using Phaser [64] and the coordinates of crystal structure of PIM-1 (PDB 2J2I; [65,66]) as the search model. Model rebuilding and structure refinement were performed in COOT [67] and REFMAC5 [68], respectively. Data collection and refinement statistics are summarized in Appendix A. For preparation of the illustrative materials (Figure 2 and Figure 3), ICM software was used (Molsoft L.L.C.; San Diego, CA, USA).

### 3.5. Biochemical Binding/Displacement Assays with Measurement of FA or TGLI

Optical FA measurement modules for 5-TAMRA-labeled compounds (ex. 540(20) nm, em. 590(20) nm and 590(20) nm), for Cy5-labeled compounds (ex. 590(50) nm, em. 675(50) nm and 675(50) nm), and optical TGLI measurement modules for ARC-1451 (ex. 330(60) nm, em. 590(50) nm), for ARC-1188 (ex. 330(60) nm, em. 675(50) nm) were used for assays with PHERAstar microplate reader (BMG Labtech; Ortenberg, Germany). TGLI measurements were performed with a delay time of 50 μs and integration of the signal for 150 μs. In each measurement, emission from 200 flashes of excitation radiation was collected. The measurement solutions (20 µL volumes) were prepared in black 384-well low-binding surface microtiter plates (Corning #4514; Kennebunk, ME, USA) and incubated for 20 min at 30 °C before the measurements. The assay buffer contained HEPES hemisodium salt (pH 7.4, 50 mM), NaCl (150 mM), dithiothreitol (5 mM), bovine serum albumin (0.5 mg/mL), and Tween-20 (0.005%). GraphPad Prism v. 5.04 (GraphPad Software, Inc.) was used for data analysis with FA [32] or TGLI [33] readouts. Representative titration curves can be found in Appendix A.

### 3.6. Kinase Selectivity Profiling

Kinase selectivity profiling was performed using the commercial service provided by the MRC PPU International Centre for Kinase Profiling (Dundee, UK) using a radiometric filter-binding assay [69]. Kinases used for analysis were of human origin. The ATP concentration applied for inhibition measurements was at or below the ATP *K_M_* value of the particular PK. ARC-3125 and ARC-3126 were screened against 139 PKs, whereas compounds **1**, **2**, and **8** were screened against 140 PKs. The final total concentration of inhibitors in the selectivity panel was 1 µM. Residual activities were expressed as the percentage of retained activity compared to the control without inhibitor.

From the residual activities, values of the Gini coefficient and hit rate were calculated for each individual compound. The Gini coefficient is a single-value metric whose value for a specific inhibitor depends on the number and origin of the PKs in the panel, but also on concentration of the tested compound [70]. Hit rate is the number of kinases inhibited by over 50%.

### 3.7. Surface Sandwich Assay for PIM-2 Measurement

The composition of the assay buffer was as follows: HEPES (50 mM), NaCl (150 mM), and Tween 20 (0.005%).

The Pierce streptavidin-coated microtiter plate (#15407; Waltham, MA, USA) was used for the preparation of the ARC-affinity surface. ARC-2073 in 10 mM phosphate buffer (pH = 7.0; 80 nM, 50 μL) was added to the wells and incubated for 60 min at room temperature (RT) under constant shaking at 300 rpm. After incubation, the wells were washed with assay buffer (100 μL) for 10 s at RT under constant shaking at 300 rpm. All the following washing procedures were performed similarly. Thereafter, PIM-2 in assay buffer (50 μL) was added to the ARC affinity wells. The wells were incubated for 60 min at RT under constant shaking at 300 rpm. After another washing procedure, primary antibody D1D2 (1.7 nM, 50 μL) in assay buffer was incubated in the wells for 60 min at RT under constant shaking at 300 rpm. The washing was repeated. Finally, europium chelate-labeled secondary antibody (G0506) (2.2 nM, 50 μL) in assay buffer was incubated in the wells for 60 min at RT under constant shaking at 300 rpm. The wells were washed, dried under a fume hood for 10 min, and TGLI was measured on their surfaces. GraphPad Prism v. 5.04 was used for data analysis. The limit of quantification (LoQ) was calculated based on the standard deviation of the calibration curve (SD) and the slope of the calibration curve (S) using the equation LoQ = 10 × SD/S.

PC-3 cells were cultured in DMEM high-glucose medium supplemented with 10% FBS, 2 mM glutamine, 100 U/mL penicillin, 100 µg/mL streptomycin, and 0.25 µg/mL amphotericin B at 37 °C in humidified atmosphere containing 5% CO_2_. PC-3 cells were suspended in assay buffer supplemented with EDTA (1 mM), protease inhibitor cocktail (1x), and phenylmethylsulfonyl fluoride (0.5 mM). The lysates were prepared by exposing PC-3 cells to continuous-wave 40-kHz ultrasound at an intensity of 0.4 W/cm^2^ for 10 s three times in ultrasonic bath at RT with a 30 s interval on ice to avoid overheating [71].

### 3.8. Cell Culture, Transfection, and Imaging

U2OS cells were cultured in McCoy’s 5A medium supplemented with 10% FBS, 2 mM glutamine, 100 U/mL penicillin, 100 µg/mL streptomycin, and 0.25 µg/mL amphotericin B at 37 °C in humidified atmosphere containing 5% CO_2_. The imaging was performed at 30 °C in humidified atmosphere containing 5% CO_2_.

For studies of the general internalization properties of ARCs, the cells were seeded at the density of 12,000–25,000 cells per well onto an Ibidi μ-plate (96-well; Ibidi GmbH, Planegg, Germany). After 24 h incubation in growth medium, the cells were treated for 10 min, 30 min, or 1 h with dilutions of compounds **6**, **7**, **8**, or ARC-2090 (5 μM, 2.5 μM, or 1 μM final concentration) in the phenol red-free DMEM/Ham’s F12 medium supplemented with 2% ccFBS and 2 mM glutamine. Next, cells were washed twice with PBS, and imaging medium (MEMO + Supplement A 1:100) was applied subsequently. Cells were imaged with Cytation 5 wide-field microscope (BioTek; Winooski, VT, USA) using 20× air objective and the following settings: for Cy5, 623 nm LED (intensity: 2) and Cy5 filter cube (685 nm) with 104 ms integration time and detector gain value of 2; for TAMRA, 523 nm LED (intensity: 7) and RFP filter cube (593 nm) with 252 ms integration time and detector gain value of 16; for bright-field, white LED with intensity 6, 186 ms integration time and detector gain value of 15. Four independent experiments were performed.

For the co-localization analysis, the cells were seeded at the density of 30,000 cells per well onto the 8-well CG imaging chambers (Zell Kontakt GmbH; Nörten-Hardenberg, Germany). After 24 h incubation in growth medium, the cells were transfected for 48 h with mixtures comprising 0.8 μg of plasmid (*mCherry-Endo-14*, *GFP-PIM-1*, *PIM-1-TagRFP,* or *mCherry-Peroxisomes-2*) and 2.4 μL of FuGENE^®^ HD per well. The following treatment of cells with compounds **6** or **7** (5 μM or 2.5 μM final concentrations) was performed as described above. Imaging was carried out with the TIRF microscopy setup as described in [72]. Briefly, widefield epifluorescence and Highly Inclined and Laminated Optical sheet (HILO) imaging [73] was conducted using an inverted microscope built around a Till iMIC body (Till Photonics/FEI; Munich, Germany), equipped with TIRF APON 60× oil (NA 1.49) objective lenses (Olympus Corp., Tokyo, Japan). The samples were sequentially excited with 488 nm (100 mW), 515 nm (80 mW), or 638 nm (150 mW) PhoxX laser diodes (Omicron-Laserage; Rodgau, Germany) combined in the SOLE-6 light engine (Omicron-Laserage). Excitation light was launched into Yanus scan head, which along with a Polytrope galvanometric mirror (Till Photonics/FEI; Munich, Germany) was used to position the laser for widefield epifluorescence or HILO illumination. Excitation and emission light was spectrally separated with imaging filter cubes consisting of a zt 491 rdc beamsplitter (Chroma) and 525/45 BrightLine emitter filter (Semrock; West Henrietta, NY, USA) for proteins labeled with GFP; zt 514/640 rpc dual line beamsplitter (Chroma) and the 577/690–25 nm BrightLine dual-band bandpass emission filter (Semrock) for mCherry or TagRFP-labeled proteins with 515 nm excitation laser or for Cy5-labeled ligands with 638 nm excitation laser. Additionally, the brightfield channel was used for the determination of cell borders and nucleus position. The electron-multiplying charge-coupled device Ultra 897 camera (Andor Technology; Belfast, UK) was mounted to a microscope through a TuCam adapter with 2× magnification (Andor Technology). The camera was cooled down to −100 °C with the assistance of a liquid recirculating chiller Oasis 160 (Solid State Cooling Systems; Wappingers Falls, NY, USA).

All measurements were performed the 8-well CG imaging chambers (Zell Kontakt GmbH), and 3 to 9 randomly selected areas per well were captured as 16-bit OME-TIFF images at EM gain 300 with 100 ms exposure time. In case of ARCs, the maximal laser power of 15% was used; in case of genetically encoded proteins, the imaging settings were varied to enable optimal signal levels (the absence of detectable crosstalk signal in the channels was ensured). Live Acquisition v.2.07 (TILL Photonics/FEI) software autofocus was used to determine the focal plane, and multicolor Z-stacks (100 frames, with 200 nm piezo-focusing increment) were taken. The z-stack was deconvolved with EpiDEMIC plugin [74], the objects were segmented with Spot Detection plugin in an ICY platform [75], and the degree of multichannel co-localizations were estimated by Mander’s coefficients using Colocalization Studio [76]. Two independent experiments were performed.

## 4. Conclusions

We report the first high-resolution structures from co-crystals of PIM-1 protein and ARC inhibitors. The X-ray crystallographic study confirmed the bisubstrate binding of ARCs to PIM kinases [14], where the adenosine analogue and peptide mimetic moieties bind to the corresponding substrate pockets of PIM-1 (Figure 2). Considering the structural similarity among PIM kinases, ARCs are also bisubstrate inhibitors of PIM-2 and PIM-3 proteins (Table 1).

New ARC inhibitors were constructed by deleting structural elements that lacked interaction with PIM-1 based on the co-crystal structures. These simplified conjugates still retained low-nanomolar affinity and remarkable selectivity toward the kinases of the PIM family.

A biotinylated ARC inhibitor was tested as a kinase capture agent for a surface sandwich assay and used in combination with a PIM-2 detection antibody and secondary antibody labeled with Eu chelate for time-gated measurement of photoluminescence. This highly sensitive approach revealed an LoQ value of 44 pg recombinant PIM-2 protein in a biochemical assay. Further optimization of the assay for its use for analysis of PIM-2 in complex biological matrices is in progress.

ARCs labeled with Cy5 red fluorescent dye were efficiently taken up by live U2OS cells. Partial co-localization of ARC probes with the endosomal marker mCherry-Endo-14 was observed. In U2OS cells transiently transfected with PIM-1 fused with a fluorescent protein, a high extent of co-localization of PIM-1 with fluorescent ARC probes was evident.

## Figures and Tables

**Figure 1 molecules-26-04353-f001:**
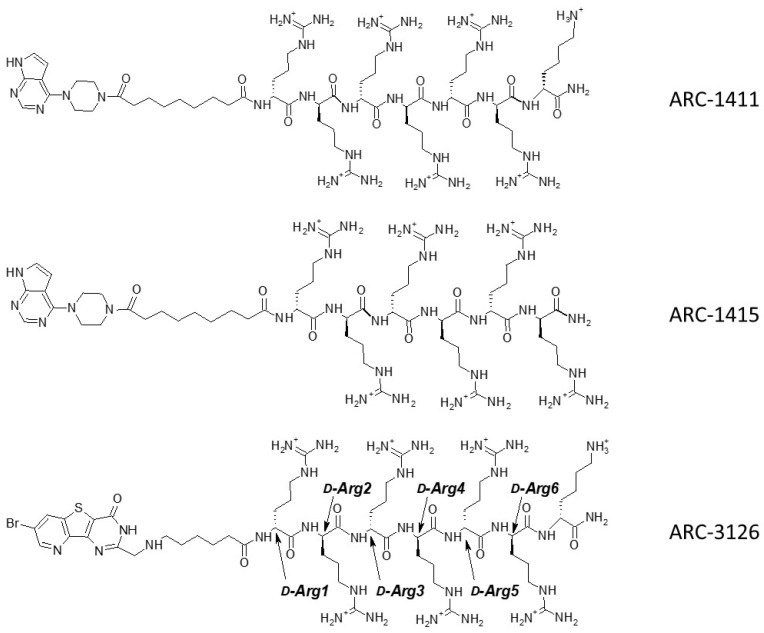
Structures of ARC inhibitors co-crystallized with PIM-1.

**Figure 2 molecules-26-04353-f002:**
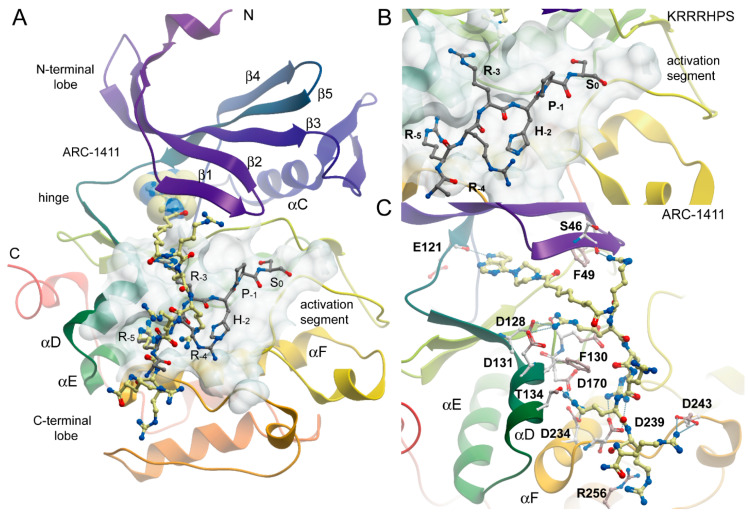
Binding mode of ARC-1411 in PIM-1 and comparison with a PIM substrate complex. (**A**) Overall structure of the PIM-1 catalytic domain in complex with ARC-1411 (PDB-ID: 7OOV). The main secondary structure elements are labeled. ARC-1411 is shown in a ball and stick representation, and the ATP mimetic fragment of the inhibitor is additionally highlighted in transparent CPK. The binding site of ARC-1411 is shown as a semi-transparent surface. The structural model has been superimposed with a PIM-1 substrate complex with the consensus peptide KRRRHPS (PDB-ID: 2BIL). The peptide is also shown in ball and stick representation with gray carbon atoms. (**B**) Details of the interaction of the PIM-1 complex with the consensus peptide. (**C**) Details of the interactions of ARC-1411 with PIM-1.

**Figure 3 molecules-26-04353-f003:**
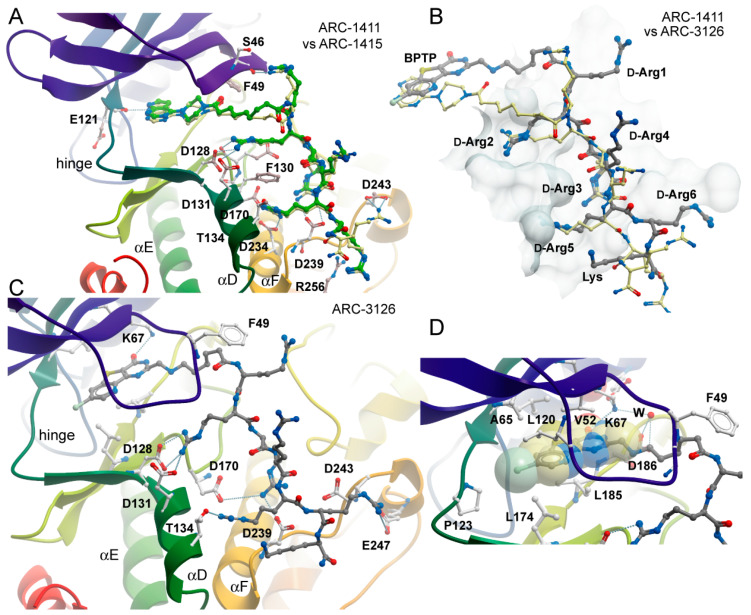
Binding modes of ARC-1415 and ARC-3126. (**A**) Comparison of the binding modes of ARC-1411 and ARC-1415. The figure shows a superimposition of both structures. ARC-1411 carbon atoms are colored in yellow, and ARC-1415 carbon atoms are colored in green. Main interacting residues of PIM-1 are labeled. (**B**) Comparison of the binding modes of ARC-1411 and ARC-3126. Both structures have been superimposed, but for clarity, only the inhibitors are shown on the surface representation of PIM-1. ARC-1411 carbon atoms are colored in yellow and ARC-3126 carbon atoms are colored in gray. The residues in ARC-3126 are labeled. (**C**) Details of the interactions formed by ARC-3126. The main interacting residues are labeled. ARC-3126 carbon atoms are colored in gray. (**D**) Details of the interactions of the ATP-mimetic BPTP moiety and the linker in the ARC-3126 complex. W stands for water molecule.

**Figure 4 molecules-26-04353-f004:**
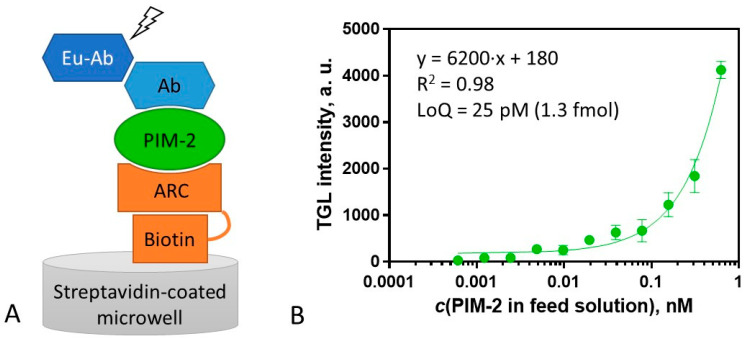
Scheme and calibration curve of the assay for measurement of concentration of PIM-2. (**A**) Scheme of the assay format for PIM-2 measurement. PIM-2 is bound to the ARC-2073 functionalized surface (step 1), the solution of the primary antibody (Ab) is added (step 2), and Eu chelate-labeled secondary antibody (Eu-Ab) is added (step 3). (**B**) Calibration curve for PIM-2 analysis (immobilized ARC-2073 (4 pmol), detection with D1D2 (84 fmol) and G0506 (110 fmol)). TGL intensity was measured with excitation at 330 (60) nm; emission was detected at 590 (50) nm (integration from 80 to 1000 μs). All measurements were performed in triplicate, and the variance was estimated as SEM.

**Figure 5 molecules-26-04353-f005:**
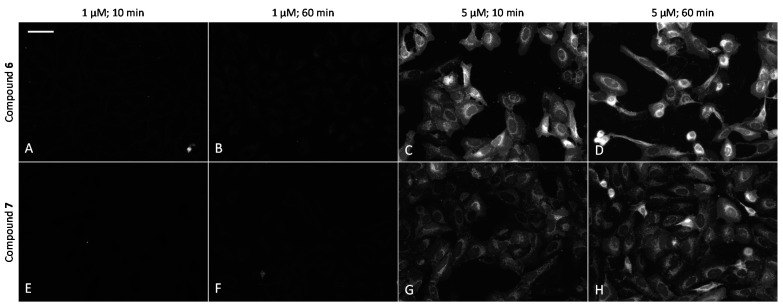
Comparison of cellular uptake of ARCs labeled with Cy5 dye into live U2OS cells. (**A**–**D**) compound **6**; (**E**–**H**) compound **7**. Incubation conditions are listed above the images. Representative images from a single experiment are shown; in all wells, imaging settings were identical (see Materials and Methods section of the main text). Scale bar: 50 μm.

**Figure 6 molecules-26-04353-f006:**
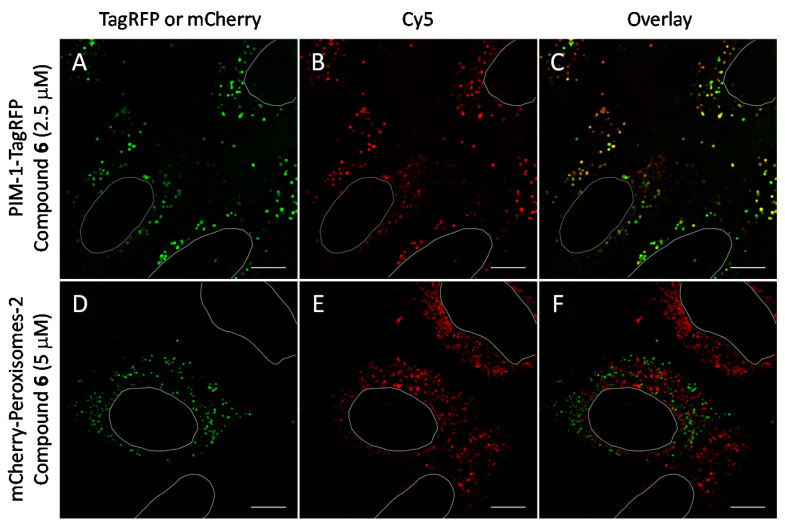
Co-localization studies of PIM-targeting compound **6** with fluorescent protein-tagged PIM-1 and peroxisomal marker (following 48 h transfection) in live U2OS cells. The PIM-1-mRFP and mCherry-Peroxisomes-2 are shown in green and compound **6** is shown in red. White ovals indicate nuclei borders. Individual channels and overlayed images: (**A**–**C**) PIM-1-TagRFP and compound **6** (2.5 μM, 1 h incubation); (**D**–**F**) mCherry-Peroxisomes-2 and compound **6** (5 μM, 1 h incubation). Representative images (in cells with low expression of constructs) from two independent experiments are shown. Scale bar: 10 μm.

**Figure 7 molecules-26-04353-f007:**
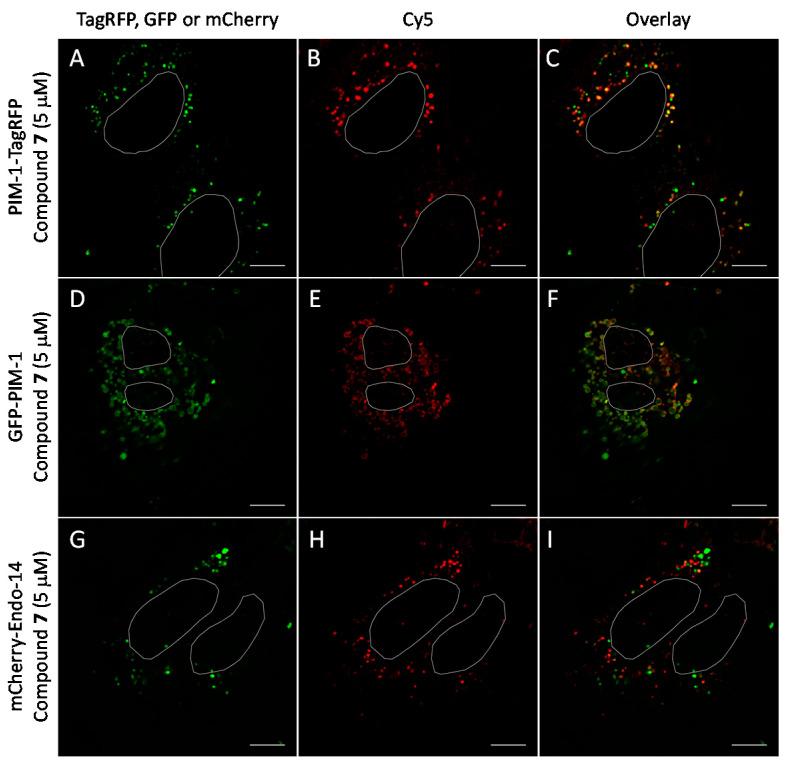
Co-localization studies of PIM-targeting compound **7** (5 μM, 1 h incubation) with fluorescent protein-tagged PIM-1 and endosomal marker mCherry-Endo-14 in live U2OS cells. The genetically encoded PIM-1 constructs are shown in green and compound **7** is shown in red. Individual channels and overlaid images: (**A**–**C**) PIM-1-TagRFP and compound **7**; (**D**–**F**) GFP-PIM-1 and compound **7**; (**G**–**I**) mCherry-Endo-14 and compound **6**. Representative images (in cells with low expression of constructs) from two independent experiments are shown. White ovals indicate nuclei borders. Scale bar: 10 μm.

**Table 1 molecules-26-04353-t001:** Values of thermal shift of screened compounds in complex with PIM kinases.

Compound	Structure	∆*T*_m_, °C		
		PIM-1	PIM-2	PIM-3
**ARC-684**	PYB-Ahx-d-Arg-Ahx-(d-Arg)_6_-d-Lys-NH_2_	7.4 ± 0.2	7.7 ± 0.5	6.3 ± 0.3
**ARC-668**	AMTH-Ahx-d-Arg-Ahx-(d-Arg)_6_-d-Lys-NH_2_	6.7 ± 0.2	8.1 ± 0.6	6.9 ± 0.3
**ARC-1141**	AMTH-Ahx-d-Ala-(d-Arg)_6_-d-Lys-Gly	5.2 ± 0.4	5.9 ± 0.6	4.9 ± 0.4
**ARC-1411**	PIPY-C(=O)-(CH_2_)_7_-C(=O)-(d-Arg)_6_-d-Lys-NH_2_	7.6 ± 0.3	8.4 ± 0.4	6.0 ± 0.2
**ARC-3119**	BBTP-Hyp-Ahx-(d-Arg)_9_-d-Lys-NH_2_	13.7 ± 0.4	13.9 ± 0.6	11.9 ± 0.2
**ARC-3125**	TIBI-CH_2_-C(=O)-Ahx-(d-Arg)_6_-d-Lys-NH_2_	12.7 ± 0.2	14.7 ± 1.0	12.9 ± 0.2
**SGI-1776**	Not shown	9.5 ± 0.4	8.1 ± 0.6	8.2 ± 0.3

AMTH-5-(2-aminopyrimidin-4-yl)-thiophene-2-carboxylic acid moiety; Ahx-6-aminohexanoic acid moiety; PYB-3-(pyridin-4-yl)benzoic acid; PIPY-4-(piperazin-1-yl)-7H-pyrrolo[2,3-d]pyrimidine moiety; TIBI-(4,5,6,7-tetraiodo-1H-benzimidazol-1-yl)acetic acid moiety; BBTP-8-bromo-2-(methylene)benzo[4,5]thieno[3,2-d]pyrimidin-4-one moiety. Mean values ± standard error of the mean (SEM) are shown (N = 2).

**Table 2 molecules-26-04353-t002:** Structures of tested ARC inhibitors and their affinities toward PIM kinases and PKAcα.

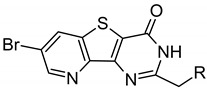 ≡ BPTP 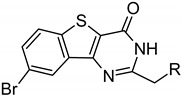 ≡ BBTP					
Compound #	Structure ^a^	*K_D_*, nM (PIM-1) ^b^	*K_D_*, nM (PIM-2) ^b^	*K_D_*, nM (PIM-3) ^b^	*K_D_*, nM (PKAcα) ^b^
**ARC-3126**	BPTP-Ahx-(d-Arg)_6_-d-Lys-NH_2_	1.8 ± 0.7	2.9 ± 0.5	2.0 ± 0.5	>18,460
**1** (ARC-2067)	BPTP-**Aoc-(d-Arg)_2_-Gly-d-Arg**-d-Lys-NH_2_	11.4 ± 1	64 ± 8	5.2 ± 1	2250 ± 411
**2** (ARC-2059)	**BBTP**-Aoc-(d-Arg)_2_-Gly-d-Arg-d-Lys-NH_2_	0.4 ± 0.1	3.8 ± 0.2	2.0 ± 0.4	853 ± 28
**3** (ARC-2060)	BBTP-**Amt**-(d-Arg)_2_-Gly-d-Arg-d-Lys-NH_2_	0.7 ± 0.2	9.9 ± 0.8	7 ± 1	1502 ± 266
**4** (ARC-2061)	BBTP-**Tap**-(d-Arg)_2_-Gly-d-Arg-d-Lys-NH_2_	1.8 ± 0.3	20 ± 4	6.6 ± 1.4	>18,210
**5** (ARC-2062)	BBTP-**Hyp-Aoc**-(d-Arg)_2_-Gly-d-Arg-d-Lys-NH_2_	37.8 ± 9	447.4 ± 52	30.4 ± 6.5	>20,590
**6** (ARC-2074)	BBTP-Aoc-(d-Arg)_2_-Gly-d-Arg-[d-Lys(**Cy5**)]-NH_2_	1.7 ± 0.4	5.2 ± 0.9	2.6 ± 0.3	>303
**7** (ARC-2076)	BBTP-**Tap**-(d-Arg)_2_-Gly-d-Arg-[d-Lys(Cy5)]-NH_2_	21.9 ± 4	28.2 ± 6.7	8.3 ± 1	>1076
**8** (ARC-2065)	BBTP-Tap-(d-Arg)_2_-Gly-d-Arg-[d-Lys(**5-TAMRA**)]-NH_2_	20.5 ± 4	157 ± 39	15.7 ± 3	>1020
PIM peptide	(d-Arg)_2_-Gly-d-Arg-d-Lys-NH_2_	>11,490	>46,020	> 16,420	>89,000
AZD1208	Not shown	5.3 ± 1	11.8 ± 2	2.3 ± 0.4	>83,000

BPTP-7-bromo-2-(methylene)pyrido[4,5]thieno[3,2-d]pyrimidin-4-one moiety; Ahx-6-aminohexanoic acid moiety; Aoc-8-aminooctanoic acid moiety; BBTP-8-bromo-2-(methylene) [1]benzothieno[3,2-d]pyrimidin-4-one moiety; Amt-[4-(aminomethyl)-1H-1,2,3-triazol-1-yl]acetic acid moiety; Tap-5-(1H-1,2,3-triazol-4-yl)pentanoic acid moiety; Hyp-hydroxyproline. ^a^ The fragments shown in bold indicate structural changes compared to the previous compound in the series. ^b^ *K_D_* values were determined in binding/displacement assays with TGLI or FA readout using probes ARC-1451, ARC-1188, or ARC-583 [32,33]. The full structures of ARCs are depicted in Appendix A. Mean values ± SEM are shown (N = 2).

## Data Availability

The crystallographic data are available from https://www.rcsb.org/ (PDB codes: 7OOV for PIM-1-ARC-1411, 7OOW for PIM-1-ARC-1415, 7OOX for PIM-1-ARC-3126). Other data are available upon reasonable request to the corresponding author.

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
