# Peer review of "Crystal Structure-Guided Design of Bisubstrate Inhibitors and Photoluminescent Probes for Protein Kinases of the PIM Family"

_molecules, 2021, doi:10.3390/molecules26144353_

Round 1

Reviewer 1 Report

The manuscript written by Olivier E. Nonga, Darja Lavogina, Erki Enkvist, Katrin Kestav, Apirat Chaikuad, Sarah E. Dixon-Clarke, Alex N. Bullock, Sergei Kopanchuk, Taavi Ivan, Ramesh Ekambaram, Kaido Viht, Stefan Knapp and Asko Uri, entitled "Crystal structure-guided design of bisubstrate inhibitors and 2 photoluminescent probes for protein kinases of the PIM family" describes organic synthesis and biochemical studies of a new designed protein kinase PIM-1 inhibitors containing three parts: adenosine mimetic moiety, linker, and an unnatural peptide chain (D-Arg)6. Additionally, three crystal structures of protein kinase (PIM-1) in complex with ARC highly potent bidentate inhibitors were determined. These complexes revealed a network of interactions with both the ATP-binding pocket and the substrate protein-binding site on the C-terminal kinase lobe of PIM-1. The advantage of these compounds is their relatively simple chemical structure with a reduced number of polar groups and chiral centers. The obtained inhibitors possess nanomolar potency and good selectivity towards PIM-1 kinases. The advantage of this research is extensive biochemical study of ARCs labelled with a fluorescent Cyanine5 dye (Cy5). Usage of this fluorescent probe allowed explanation of PIM  function in cellular systems in normal and disease physiology. 

In its current form the paper is dedicated to the narrow group of researchers working on kinase PIM inhibitors. The authors should try to make this paper more intelligible for a wider group of readers. I would suggest:

  1. The abstract should be shortened and more focused on the main goal of the paper. The authors have to decide what “story” they really want to tell .
  2. Crystallographic studies create a crucial part of the work, so, the interactions of PIM-1 with inhibitors should be described in more detail,
  3. Figure 2 could be improved and the electron density maps covering inhibitors should be provided.
  4. In the main text of the manuscript should be shown structural formulas of inhibitors ( ARC-1411, -1415 and -3126) used for PIM-1 complex crystal structures.

There are some errors with citations.

  1. The supplementary material is overloaded and unordered. More explanation is needed.

- Table S1 could be removed, all compounds according to the citation were described in [1-6].

- Table S2. In this table structural formulas (not structures) are shown of the newly synthesized inhibitors. The title of this table should be corrected.

- All figures S1-S6 need some comments and conclusions from the presented data.

Author Response

We thank the reviewers for the feedback regarding our manuscript. Below, we provide point-by-point response to the issues raised by the reviewers.

  1. We re-wrote the abstract, please see lines 19-29 (tracked changes version: lines 19-47) of the main text.
  2. We expanded the discussion of the new crystal structures, please see lines 119-195 (tracked changes version: lines 140-228) of the main text.
  3. We changed Figure 2: in the new version, we are showing the overlay of co-crystals containing ARC-1411 and ARC-1415 (part A), or ARC-1411 and ARC-3126 (part B), and in detail the interactions between ARC-3126 and PIM-1 (parts C-D). Please note that the numbering of the Figure in the main text has now changed (to Figure 3) due to introduction of an additional Figure according to your following comments. Additionally, we provided the electron densities resolved for the ligand part in each crystal structure as Supplementary Figure S1.
  4. We introduced Figure 1 with formulas of ARC-1411, ARC-1415, and ARC-3126.
  5. We deleted Table S1. We changed the title of the Supplementary Table as requested. Please note that the numbering of the Supplementary Table has now changed (to Table S1). As requested, we added some explanatory sentences to the legends of Supplementary Figures (S2-S7, according to the new numbering).

Reviewer 2 Report

Nonga et al. determined crystal structures of PIM in complexed with bisubstate inhibitors, and performed the biochemical experiments and cell biology experiments. The comprehensive research information of this manuscript is considered to be a contribution to the relevant field. I support the publication of this article after solving the some concerns as below,

1. I found many "Error! Reference source not found.." in the manuscript. Correction is required.

2. "PDB codes: 70OV for PIM1-ARC-1411, 70OW for PIM1-ARC-1415, 70OX for PIM1-ARC-3126" I searched the PDB to check the electron density map, but I couldn't access it with the pdb code written in the manuscript. Authors should contact the PDB to resolve this. If the deposited data is held before publication, it is suggested to submit a PDB validation report. Also I suggest adding electron density maps for inhibitors for reliability of the data.

3. The content of the abstract is too comprehensive and detailed. It is necessary to write concisely according to the number of words suggested by the journal.

Minor

1. line 504-506: "Chemicals and resins for synthesis were obtained from Iris Biotech, Novabiochem, 504 AnaSpec, ABCR, Acros, Alfa Aesar, Fluka, Macherey-Nagel, and Fisher. SGI-1776 and 505 AZD1208 were from Axon Medchem and Cayman Chemicals, respectively." Authors need to add information about the company (region, country).

2. line 527: "...Addgene (plasmid # 54520 and # 55040, respectively).." Remove one period.

3. line 563-564: "published coordinates of PIM1 [65,66]." should be "coordinate of crystal structure of PMM1 (PDB code; XXXX) as the search model"

4. Supplementary information: Author's name and affliation should be added.

5. Figure 3A: The text overlaps the shape. It needs to be corrected for publication.

Author Response

We thank the reviewers for the feedback regarding our manuscript. Below, we provide point-by-point response to the issues raised by the reviewers.

Major issues:

  1. We are sorry that the version of the main text where the field codes were not removed contained error messages due to incompatibility with the submission system. This time, we submitted the main text with field codes removed, and we hope that this will solve the aforementioned issue.
  2. We are adding PDB validation report as the supplementary material (3 PDF-files for the review purposes). Furthermore, we have now made public the corresponding entries in the PDB.
  3. We re-wrote the abstract, please see lines 19-29 (tracked changes version: lines 19-47) of the main text.

Minor issues:

  1. We added the regions and countries for the chemicals, materials, and apparatuses used throughout the Materials and methods part of the main text, please see lines 450-575 (tracked changes version: lines 484-612).
  2. The extra period was removed, please see line 475 (tracked changes version: line 509) of the main text.
  3. The requested change was introduced, please see lines 508-509 (tracked changes version: lines 544-545) of the main text.
  4. The requested change was introduced.
  5. The requested change was introduced. Please note that the numbering of the Figure has now changed (to Figure 4A) due to introduction of an additional Figure according to the request by another reviewer.

Round 2

Reviewer 2 Report

The author addressed the reviewers' concerns about the original manuscript Meanwhile, I request some revisions to the revised manuscript.

-Authors should write a manuscript according to the journal format.

Supplementary Data
-For supplementary data, yellow shades should be excluded.

In Table S1. 
- "Rfact" should be "Rfactor"
- "22% PEG3350" and "15% PEG 10000" should be "22% (w/v) PEG 3350" and "15% (w/v) PEG 10000", respectively.
- "tris" should be "Tris" or "Tris-HCl"